# Damage Localization of Beam Bridges Using Quasi-Static Strain Influence Lines Based on the BOTDA Technique

**DOI:** 10.3390/s18124446

**Published:** 2018-12-15

**Authors:** Yang Liu, Shaoyi Zhang

**Affiliations:** School of Transportation Science and Engineering, Harbin Institute of Technology, Harbin 150090, China; zhangshaoyi@outlook.com

**Keywords:** damage localization, beam bridges, quasi-static strain influence line, BOTDA technique

## Abstract

The diagnosis of damage in a bridge superstructure using quasi-static strain influence lines (ILs) is promising. However, it is challenging to accurately localize the damage in a bridge superstructure due to limited numbers of strain IL measurement points and inconsistencies between the loading conditions before and after damage. To address the above issues, the Brillouin optical time domain analysis (BOTDA) technique is first applied to bridge damage localization using quasi-static strain ILs, and the number of strain IL measurement points is substantially increased. Additionally, a damage localization index based on quasi-static strain ILs that is independent of differences in the loading conditions before and after damage is proposed to localize damage in the superstructure of a beam bridge. Finally, the effectiveness of the proposed method is demonstrated through both numerical analysis and measured data from a quasi-static test of a model bridge.

## 1. Introduction

The strain of structures is generally recognized as a highly sensitive feature in the damage of civil structures [1]. With the progress of technology of strain sensors [2,3], methods based on strain information have been paid much attention in the damage diagnosis of civil structures, and thus, some damage diagnosis indices using strain of structures are investigated, such as the strain modal analysis [4,5,6], the modal strain energy [7,8,9,10], and the strain data [11,12] etc. In contrast to dynamic methods, static methods are effective for not only ensuring that measured data boast a high accuracy but also avoiding the effects of the mass and damping of structures [13,14]. Furthermore, during the operational period, it is relatively easy to measure the static deformation of actual bridges, e.g., the displacement and strain of the bridge superstructure, through the static load test. Therefore, some researchers have focused on diagnosing damage using the static deformation of structures [15,16,17]. Among all the static features available for this purpose, the strain/displacement influence line (IL) is believed to be a particularly promising index to diagnose and detect structural damage in actual bridges [18]. Thus, methods based on the difference between ILs before and after damage have been proposed to both localize and detect damage in bridges [19,20].

For operational bridges, theoretical static IL cannot be obtained; instead, only a quasi-static moving load, i.e., a vehicle driving with a slow and uniform velocity, can be implemented to generate the ILs of the bridge superstructure. Under the above quasi-static loading condition, the strain/displacement IL is used to approximate the static IL; hence, this type of real IL is known as the quasi-static IL [21,22,23]. Taking advantage of the characteristics of quasi-static strain ILs, their application to the localization of damage within bridges has been investigated to some degree. Under a quasi-static moving load, a method based on stress ILs was implemented to localize the damage in an actual suspension bridge [24]. Quasi-static strain ILs were also used to localize the damage in bridges without a baseline finite element model, and the feasibility, effectiveness and limitations of this approach were investigated both numerically and experimentally [25]. Furthermore, the relationship between the damage region and the area of the strain IL was investigated and subsequently used to localize the damage in a continuous beam bridge [26].

Although quasi-static strain ILs are effective for locating the damage of bridges in theory, as discussed above, the limited number of measured strain ILs impedes their application to actual bridges. Strain is known to reflect the local deformation of structures, and thus, traditional strain gauges cannot detect damage effectively unless they are located directly within the damaged region. To overcome this issue, the Brillouin optical time domain analysis (BOTDA) technique [27,28] is combined with quasi-static strain ILs to supply a sufficient number of measured strain IL points. Unlike those employed in traditional strain measurement techniques, sensors based on the BOTDA technique measure the strain along the entire length of the optical fiber with a measurement range of approximately 100 km and an overall spatial resolution of measured points of approximately 1 cm [29,30,31]. Using this high spatial resolution, the BOTDA technique has been successfully applied to the detection of crack damage in bridges [32,33,34]. Similarly, if one single optical fiber is placed along the longitudinal direction of the superstructure of the beam bridge under quasi-static loading conditions, the combination of this sensing optical fiber and the BOTDA technique can supply a sufficient number of quasi-static ILs at different measured points. Theoretically, this optical fiber enables the acquisition of multiple types of damage information across the entire bridge superstructure (e.g., beam body damage, concrete crack, concrete spalling etc.) as long as the damage induces a difference between the quasi-static strain ILs before and after the damage is experienced. Therefore, the BOTDA technique can be applied to localize the damage of actual beam bridges by effectively expanding the number of measured quasi-static strain ILs.

Two additional key factors also influence the localization of damage within bridge superstructures using quasi-static strain ILs. On the one hand, it is difficult to avoid inconsistencies between the quasi-static loading conditions before and after damage during actual load tests on bridges. On the other hand, it is important to minimize the effects of measurement noise on the quasi-static strain ILs during load tests on actual bridges. The heavy influence of measurement noise can commonly produce misleading damage localization results. To address the above issues, a damage localization index based on quasi-static strain ILs is proposed that is independent of the difference between the weight of the loading vehicle before and after the bridge suffers damage. Additionally, a method based on the energy ratio between the main components of the strain IL and measurement noise is implemented to mitigate the impacts of measurement noise on the damage localization results.

Thus, the remainder of this paper is organized as follows. The damage localization approach based on quasi-static strain ILs is introduced in detail in the next section. The performance of the proposed method is discussed numerically in Section 3. Then, the effectiveness of the proposed method is identified by using the measured data of a model bridge. Finally, conclusions are drawn.

## 2. Approach for Localizing the Damage of Bridges Using Quasi-Static Strain ILs

### 2.1. Obtaining the Quasi-Static Strain ILs of Bridges Based on the BOTDA Technique

#### 2.1.1. Generating the Quasi-Static Strain ILs of Bridges

Theoretically, if the frequency band of the loading vehicle excitation is lower than the first natural frequency of a bridge, the condition of the bridge is infinitely close to the static situation. To achieve this loading condition, the loading vehicle is usually forced to drive over the bridge along one lane at a constant slow velocity after closing the bridge to traffic. The load described above is known as the quasi-static load. Although the loading vehicle has more than one axis along the longitudinal direction of the vehicle, it can be treated as one concentrated vertical load. Therefore, if one strain sensor is placed on the superstructure of the bridge, the quasi-static strain IL is obtained when the quasi-static moving load passes over the bridge.

The weight of the quasi-static moving vehicle should be sufficiently heavy to ensure that the measured strain is sensitive to the potential damage of the bridge. For actual bridges, especially existing bridges in poor condition, an overly heavy loading vehicle may induce damage. Therefore, the weight of the loading vehicle should be determined based on the real condition of the bridge on a case-by-case basis. For an actual bridge, one recommendation is to first make the reference weight of the loading vehicle equal to the weight of one loading vehicle determined by using the design of the transitional load test; then, this reference weight can be adjusted to the final loading by evaluating the real structural condition of the bridge. 

During the life cycle of operational bridges, the structural characteristics of bridge always are influenced by the variation of the environmental temperature [35,36]. Different with the long-term health monitoring of bridges, the quasi-static load test is implemented periodically, and thus, the application condition of the quasi-static load test is under control. The limited condition of quasi-static load test for actual bridges is to keep the same environmental temperature condition for each quasi-static load test of one bridge. With this way, it is effective to mitigate the influence of environmental temperature on the results obtained by the quasi-static load test. Additionally, because the quasi-static moving vehicle passes over the bridge at a uniform low speed, it is reasonable to ignore fluctuations in the speed before and after the damage. Based on the above two conditions, a method is proposed and described in detail in Section 2.2.

#### 2.1.2. Acquiring Measurements of Bridge Strain Using the BOTDA Technique

In this study, under quasi-static loading conditions, the BOTDA technique is used to obtain quasi-static strain ILs with a high spatial resolution of measured points. In the example application of the BOTDA technique shown in Figure 1, one long sensor fiber is attached along the bottom of the superstructure of a simply supported bridge.

Two laser pulses are input at each end of the fiber; one is termed the pump light, and the other is termed the probe light. Brillouin scattering inevitably occurs as the light pulse travels along the fiber. The Stokes light generated during Brillouin scattering has a frequency shift relative to the pump light [37], and this frequency shift is linearly related to the longitudinal fiber strain, as shown in the following equation.
(1)$$f_{B,i}=\;f_{B\tau ,i}\left(1+\beta \varepsilon_{i}\left(x\right)\right),$$where $f_{B,i}$ is the Brillouin frequency of the optical fiber at the *i*th measured point after deformation; $f_{B\tau ,i}$ is the reference Brillouin frequency of the optical fiber at the *i*th measured point; β is the strain coefficient with a value of 0.04812 MHz/με in this study; and $\varepsilon_{i}\left(x\right)$ is the bottom strain of the superstructure at the *i*th cross-section. Based on the above equation, the strain is calculated with the following equation.
(2)$$\varepsilon_{i}\left(x\right)=\frac{f_{B\tau ,i}-f_{B,i}}{\beta f_{B\tau ,i}}.$$

#### 2.1.3. Mitigating the Effects of Measurement Noise on the Quasi-Static Strain ILs of Bridges

In applications involving actual bridges, the measured quasi-static strain ILs are inevitably contaminated with measurement noise. To address this issue, a method based on the energy ratio between the main components of all strain ILs and the measurement noise is implemented to mitigate the effects of measurement noise. The effectiveness of this method is demonstrated by the subsequent numerical (Section 3) and experimental (Section 4) examples.

As described in Figure 1, assuming that *n* measured points are supplied by the long-distance sensing fiber, the following matrix of quasi-static strain ILs can be defined.
(3)$$\Gamma ={\left[\begin{array}{cccccc} \varepsilon_{11} & \varepsilon_{12} & \cdots & \varepsilon_{1i} & \cdots & \varepsilon_{1n}\\ \varepsilon_{21} & \varepsilon_{22} & \cdots & \varepsilon_{2i} & \cdots & \varepsilon_{2n}\\ \vdots & \vdots & \vdots & & \vdots & \vdots \\ \varepsilon_{m1} & \varepsilon_{m2} & \cdots & \varepsilon_{mi} & \cdots & \varepsilon_{mn} \end{array}\right]}_{m\times n},$$where Γ is the set of all measured strain ILs of one quasi-static load test; each column of matrix Γ is one strain IL at the *i*th measured point, i.e., the vector $\varepsilon_{i}={\left\{\varepsilon_{1i},\varepsilon_{2i},\cdots ,\varepsilon_{mi}\right\}}^{T},\left(i=1,2,\cdots ,n\right)$; and *m* is the number of all components of one strain IL. Using singular value decomposition (SVD), the matrix Γ is transformed into the following equation.
(4)$$\Gamma ={\mathrm{USV}}^{T}=\sum_{i=1}^{n}\sigma_{i}u_{i}v_{i}{}^{T},$$where U and V are the left and right singular matrixes, respectively, i.e., $U={\left[u_{1},u_{2},\cdots ,u_{i},\cdots ,u_{m}\right]}_{m\times m}$ and $V={\left[v_{1},v_{2},\cdots ,v_{i},\cdots ,v_{n}\right]}_{n\times n}$, and the matrix **S** is the singular value matrix defined as the following equation.
(5)$$S={\left[\begin{array}{c} \Sigma_{n}\\ 0 \end{array}\right]}_{m\times n},$$where $\Sigma_{n}$ is described according to the following equation.
(6)$$\Sigma_{n}=diag\left(\sigma_{1},\sigma_{2},\cdots ,\sigma_{i},\cdots ,\sigma_{n}\right),$$where diag(·) represents the diagonal matrix and $\sigma_{i}$ is the *i*th singular value $\left(\sigma_{1}>\sigma_{2}>\cdots >\sigma_{n}\right)$.

With the singular values of the strain IL matrix, the energy of all measured quasi-static strain ILs is defined by the following equation.
(7)$${\left\|\Gamma \right\|}^{2}=\sum_{i=1}^{n}\sigma_{i}{}^{2},$$As described in Equation (7), the whole energy of the strain IL matrix is evaluated by using the quadratic summation of all singular values. All the singular values are arranged from large to small; thus, the singular values with relatively large values represent the main components of the measured quasi-static strain ILs of bridges. The other singular values with small values are related to the measurement noise. Therefore, the measurement noise can be mitigated by reconstructing the matrix Γ, i.e., regrouping the matrix with all the main components. The reconstructed matrix is defined as the following equation.
(8)$$\Gamma^{\prime }=US^{\prime }V^{T}=U\left[\begin{array}{cc} \Sigma_{\tau } & 0\\ 0 & 0 \end{array}\right]V^{T},$$where $S^{\prime }$ is the reconstructed singular matrix, which takes the first τ main components from all the singular values.
(9)$$\Sigma_{\tau }=diag(\sigma_{1},\sigma_{2},\cdots ,\sigma_{\tau }),$$where τ is the number of main components of all singular values; this value is determined by satisfying the following rule.
(10)$$\xi =\frac{\left\|\Sigma_{\tau }\right\|}{\left\|\Sigma_{n}\right\|-\left\|\Sigma_{\tau }\right\|}\times 100\%\geq 0.99,$$where ξ is the energy ratio between the main components of the measurement noise in all the measured quasi-static strain ILs and 0.99 is the cut-off value, which means that 99 percent of the whole energy of matrix Γ is taken for the reconstructed matrix $\Gamma^{\prime }$. Of course, the cut-off value varies with different noise levels for actual bridges. The cut-off value should be determined by using the signal to noise ratio of measured strain ILs, but the real signal to noise ratio cannot be obtained. Therefore, this cut-off value always is determined by practical experience [38]. Based on our practical experience, it is advisable to use a value of approximately 0.99 to retain most of the information regarding all the measured quasi-static strain ILs of bridges.

### 2.2. Localizing Damage Using Quasi-Static Strain ILs Based on the BOTDA Technique

#### 2.2.1. Damage Features Based on Quasi-Static Strain ILs

As shown in Figure 1, under quasi-static loading conditions, the quasi-static strain ILs at different positions are obtained using one long optical fiber placed along the bottom of the beam of a simply supported bridge. Theoretically, the strain IL of the bridge superstructure describes the vertical deformation of a certain cross-section due to a unit load acting at different positions along the longitudinal direction of the bridge. The unit load is replaced by a certain load for practical applications. The measured quasi-static strain ILs of the bridge superstructure directly reflect the stiffness performance of the bridge; thus, quasi-static strain ILs are deemed an effective feature with respect to the damage of the bridge superstructure.
(11)$$\kappa_{i}=-\frac{M_{i}}{EI_{i}}=-\frac{\varepsilon_{i}}{h},$$where $\kappa_{i}$, $M_{i}$, $I_{i}$, and $\varepsilon_{i}$ are the curvature, bending moment, moment of inertia, and strain, respectively, of the *i*th cross-section of the beam; *h* is the distance between the strain measurement point and the neutral axis (because the whole optical fiber lies along the bottom of the beam in the longitudinal direction of the bridge, the value of *h* is assumed to be identical at all the measured points); and E is the Young’s modulus of the material. According to the Euler beam theory, the curvature can be approximated with the following definition.
(12)$$\kappa_{i}=\frac{d^{2}w_{i}}{dx^{2}}=\frac{w_{i+1}-2w_{i}+w_{i-1}}{\Delta l^{2}},$$where $w_{i-1}$, $w_{i}$, and $w_{i+1}$ represent the vertical displacements at the (*i* − 1)th, *i*th, and (*i* + 1)th measured points, respectively, and Δl is the spatial resolution determined by the BOTDA technique. With Equations (11) and (12), the strain at the *i*th measured point is calculated by the following equation.
(13)$$\varepsilon_{i}\left(x\right)=-\kappa_{i}\left(x\right)h=-\frac{h}{\Delta l^{2}}\left[w_{i+1}\left(x\right)-2w_{i}\left(x\right)+w_{i-1}\left(x\right)\right].$$

Following the virtual work principle, under the action of a quasi-static moving load *F_R_*, the displacement IL at the *i*th measured point is obtained by the following equation.
(14)$$w_{i}\left(x_{R}\right)=F_{R}\int_{0}^{l}\frac{1}{EI\left(x\right)}M_{i}\left(x\right)M_{R}\left(x\right)dx,$$where *l* is the total length of the beam and $x_{R}$ is the distance between the point being acted upon by the load *F_R_* and the supported end on the left. Furthermore, the damage is assumed to have occurred in the region around cross-section C, as shown in Figure 1, and the stiffness of the damage region is defined as the following equation.
(15)$$EI\left(x\right)=\{\begin{array}{cc} EI_{d}, & l_{c}-c\leq x\leq l_{c}+c\\ EI, & \mathrm{else} \end{array},$$where *c* is applied to describe the damaged region. the bending moment at the *i*th strain measurement point $M_{i}\left(x\right)$ is obtained by the following equation.
(16)$$M_{i}\left(x\right)=\{\begin{array}{cc} \frac{l-l_{i}}{l}x, & 0\leq x\leq l_{i}\\ -\frac{l_{i}}{l}x+l_{i}, & l_{i}\leq x\leq l \end{array},$$where $l_{i}$ is the distance between the *i*th measured point and the supported end on the left. The bending moment in Equation (14) is calculated by the following equation
(17)$$M_{R}\left(x\right)=\{\begin{array}{cc} \frac{l-x_{R}}{l}x, & 0\leq x\leq x_{R}\\ -\frac{x_{R}}{l}x+x_{R}, & x_{R}\leq x\leq l \end{array}.$$From Equation (13) to Equation (17), the quasi-static strain IL at the *i*th measured point is obtained by the following equation.
(18)$$\varepsilon_{i}\left(x\right)=-F_{R}\frac{h}{\Delta l^{2}}\left(\begin{array}{l} \int_{0}^{l}\frac{1}{EI\left(x\right)}M_{i+1}\left(x\right)M_{R}\left(x\right)dx-2\int_{0}^{l}\frac{1}{EI\left(x\right)}M_{i}\left(x\right)M_{R}\left(x\right)dx\\ +\int_{0}^{l}\frac{1}{EI\left(x\right)}M_{i-1}\left(x\right)M_{R}\left(x\right)dx \end{array}\right).$$

Based on Equations (15) and (18), the difference between the quasi-static strain ILs at the *i*th measured point before and after damage is obtained using the following equation.
(19)$$\Delta \varepsilon_{i}\left(x\right)=\varepsilon_{i,U}\left(x\right)-\varepsilon_{i,D}\left(x\right),$$where $\varepsilon_{i,U}\left(x\right)$ and $\varepsilon_{i,D}\left(x\right)$ represent the quasi-static strain ILs at the *i*th measured point before and after damage, respectively. The difference between quasi-static strain ILs before and after damage changes to the following equation.
(20)$$\Delta \varepsilon_{i}\left(x\right)=\varepsilon_{i,U}\left(x\right)-{\varepsilon^{\prime }}_{i,D}\left(x\right)=\varepsilon_{i,U}\left(x\right)-\alpha \varepsilon_{i,D}\left(x\right).$$As described in Equation (20), under inconsistent quasi-static loading conditions before and after damage, the difference between quasi-static strain ILs $\Delta \varepsilon_{i}\left(x\right)$ is determined not only by the damage of the bridge but also by the extent of the disturbance in the loading conditions before and after damage α. Therefore, an inconsistency in the loading condition produces misleading damage localization results.

To address the issue discussed above, the subspace between the quasi-static strain ILs at two different measured points is proposed to represent the damage feature for the subsequent localization of damage and is defined as the following equation.
(21)$$\rho_{i,k}=\frac{\varepsilon_{i,U}{\left(x\right)}^{T}\varepsilon_{k,D}\left(x\right)}{\left\|\varepsilon_{i,U}\left(x\right)\right\|\left\|\varepsilon_{k,D}\left(x\right)\right\|},\;i\in \left(1,\;2,\;\cdots ,\;n\right),k\in \left(1,\;2,\;\cdots ,\;n\right),$$where $\rho_{i}$ is the subspace between the quasi-static strain ILs at the *i*th measured points before and after damage, and the operation for calculating the norm of the matrix or vector is defined by the following equation.
(22)$$\left\|\varepsilon_{i,U}\left(x\right)\right\|=\sqrt{\left(\varepsilon_{1i,U}{}^{2}+\varepsilon_{2i,U}{}^{2}+\cdots +\varepsilon_{mi,U}{}^{2}\right)}.$$When the loading conditions before and after damage are inconsistent, the damage feature is calculated by the following equation.
(23)$$\begin{array}{ll} {\rho^{\prime }}_{i,k} & =\frac{\varepsilon_{i,U}{\left(x\right)}^{T}{\varepsilon^{\prime }}_{k,D}\left(x\right)}{\left\|\varepsilon_{i,U}\left(x\right)\right\|\left\|{\varepsilon^{\prime }}_{k,D}\left(x\right)\right\|}=\frac{\varepsilon_{i,U}{\left(x\right)}^{T}\left(\alpha \varepsilon_{k,D}\left(x\right)\right)}{\left\|\varepsilon_{i,U}\left(x\right)\right\|\left(\alpha \left\|\varepsilon_{k,D}\left(x\right)\right\|\right)}\\ & =\rho_{i,k} \end{array}$$The above equation demonstrates that the proposed damage feature is independent of the variation in the loading conditions before and after damage; thus, this feature is more sensitive to the damage of the bridge superstructure than to the differences between the quasi-static strain ILs before and after damage occurs. Although the proposed damage feature is not sensitive to inconsistencies in the loading conditions before and after damage, the weight of the loading vehicle before and after damage should be sufficiently heavy to obtain accurate data regarding the damage suffered by the bridge.

#### 2.2.2. Damage Localization Index and Threshold for the Determination of Damage

In this section, a Hankel matrix consisting of the proposed damage feature is established for the reference state of the bridge. A damage localization index is proposed by using the null space of this Hankel matrix, which is described in detail as follows.

For the reference state of the bridge, the matrix $\Gamma^{\prime }$ described in Equation (9) is generated by using the measured quasi-static strain ILs from one load test. Using Equation (21), a total of *s* damage features exist between an arbitrary pair of quasi-static strain ILs, i.e., ${\left\{\rho_{1,1},\rho_{1,2},\cdots ,\rho_{i,k},\cdots \right\}}_{s\times 1}$, where *s* is calculated by the following equation.
(24)$$s=\frac{n!}{2!\left(n-2\right)!},$$where ! is the factorial operator. For the quasi-static strain IL at the *i*th measured point, the following Hankel matrix is generated by using all the damage features.
(25)$$\Pi_{i,r}={\left[\begin{array}{cccc} \rho_{i,1} & \rho_{i,2} & \cdots & \rho_{i,p}\\ \rho_{i,2} & \rho_{i,3} & \cdots & \rho_{i,p+1}\\ \vdots & \vdots & \ddots & \vdots \\ \rho_{i,p} & \rho_{i,p+1} & \cdots & \rho_{i,p+q-1} \end{array}\right]}_{p\times q},$$where *p* is the number of rows in the matrix $\Pi_{i,r}$ and *q* is the number of columns in the matrix $\Pi_{i,r}$; these two numbers satisfy the following relationship.
(26)s=p+q−1,(p<q).

Based on the feature design of the Hankel matrix, each column of the matrix $\Pi_{i,r}$ is highly correlated, and each column of the Hankel matrix is known as a delay vector. For a certain Hankel matrix, *q* increases if *p* decreases, thereby reducing the degree of correlation between two delay vectors. Similarly, as the length of the delay vector *p* increases, the number of delay vectors decreases, and thus, the degree of correlation between two delay vectors decreases. Based on the basic idea of the proposed method described as follows, it is recommended to make the correlation between two arbitrary delay vectors sufficiently large. Thus, the form of the Hankel matrix should be approximately square, i.e., the value of *p* should be equal to either *q*−1 or *q*−2.

With the generated Hankel matrix $\Pi_{i,r}$, its null space satisfies the following equation.
(27)$$\Pi_{i,r}N_{i,r}=0,$$where $N_{i,r}$ is the right null space vector of $\Pi_{i,r}$ and is defined by the following equation.
(28)$$N_{i,r}=column\left(null\left(\Pi_{i,r}\right)\right),$$where null(·) is the operator of taking one column of the null space matrix, and column(·) is the operator of taking an arbitrary column of one matrix.

For the healthy state of the bridge, the damage localization index is defined by the following equation using the null space vector $N_{i,r}$ [39,40,41].
(29)$$\zeta_{i}=\Pi_{i,h}N_{i,r},$$where $\zeta_{i}\in R^{p\times 1}$ is the damage localization index. Following Equation (25), the Hankel matrix $\Pi_{i,h}$ for the healthy state of the bridge is generated by the following equation.
(30)$$\Pi_{i,h}={\left[\begin{array}{cccc} \rho_{ih,1} & \rho_{ih,2} & \cdots & \rho_{ih,p}\\ \rho_{ih,2} & \rho_{ih,3} & \cdots & \rho_{ih,p+1}\\ \vdots & \vdots & \ddots & \vdots \\ \rho_{ih,p} & \rho_{ih,p+1} & \cdots & \rho_{ih,p+q-1} \end{array}\right]}_{p\times q},$$where $\rho_{ih,p+1}$ is the subspace between $\varepsilon_{i,h}\left(x\right)$ (the healthy state) and $\varepsilon_{p+1,r}\left(x\right)$ (the reference state). The vector $\zeta_{i}$ is close to the zero vector if there is no damage within the region controlled by the quasi-static strain IL at the *i*th measured point. However, this vector cannot be perfectly zero owing to the influence of measurement noise. To evaluate this index, the following metric is defined.
(31)$$\eta_{i}=\left\|\zeta_{i}\right\|=\sqrt{\zeta_{i}^{T}\cdot \zeta_{i}}.$$

If the load test is repeated *Q* times on a bridge in healthy condition, the threshold for the damage determination is defined by the following equation.
(32)$$T_{i}=\xi_{i}\frac{1}{Q}\sum_{v=1}^{Q}\eta_{i,v},\left(v=1,2,\cdots ,Q\right),$$where $\xi_{i}$ is the adjustment factor which depends on the confidence level of the measured quasi-static strain ILs under the healthy state of bridge, and one way recommended to calculate its value is that one plus the difference between the maximum value and the minimum value of $\eta_{i}$ obtained by using the measured quasi-static strain ILs obtained at the healthy state. 

For the damaged state of the bridge (the state corresponding to the inquiry as to whether the bridge is damaged), the Hankel matrix $\Pi_{i,d}$ is generated by the following equation.
(33)$$\Pi_{i,d}={\left[\begin{array}{cccc} \rho_{id,1} & \rho_{id,2} & \cdots & \rho_{id,p}\\ \rho_{id,2} & \rho_{id,3} & \cdots & \rho_{id,p+1}\\ \vdots & \vdots & \ddots & \vdots \\ \rho_{id,p} & \rho_{id,p+1} & \cdots & \rho_{id,p+q-1} \end{array}\right]}_{p\times q},$$where $\rho_{id,p+1}$ is the subspace between $\varepsilon_{i,d}\left(x\right)$ (the state corresponding to the inquiry as to whether the bridge is damaged) and $\varepsilon_{p+1,r}\left(x\right)$ (the reference state). The damage localization index of the *i*th measured point $\zeta_{i}^{*}$ is calculated using the following equation.
(34)$$\zeta_{i}^{*}=\Pi_{i,d}N_{i,r}.$$

Then, Equation (31) is applied to obtain an estimate of the damage $\eta_{i}^{*}$. If the value of $\eta_{i}^{*}$ is larger than the threshold $T_{i}$, the region associated with the quasi-static strain IL at the *i*th measured point is considered damaged. Repeating the above procedure for all quasi-static strain ILs, all possible damaged regions can be identified. As described in Equation (25), if one quasi-static strain IL changes due to the occurrence of damage, the correlation between two arbitrary delay vectors of the whole matrix $\Pi_{i,r}$ will either increase or decrease. The larger the change in the correlation among the whole Hankel matrix $\Pi_{i,r}$, the more sensitive the damage localization index $\zeta_{i}^{}$ is to the damage. Therefore, as mentioned above, the Hankel matrix defined in Equation (25) should be approximated as a square matrix as much as possible.

### 2.3. Procedure of the Proposed Method

Based on the above description of the proposed method, the whole framework of the algorithm considers three main states: the reference state of the bridge, the healthy state of the bridge, and the damaged state of the bridge (the state corresponding to the inquiry as to whether the bridge is damaged). Usually, the first quasi-static load test is deemed the reference state of the bridge, after which the Hankel matrix $\Pi_{i,r}$ and its null space $N_{i,r}$ are determined by using the measured quasi-static strain ILs obtained in this state. The healthy state is defined as the conditions under which the bridge is believed to be in good health. For the healthy state, the damage localization index $\zeta_{i}$ and the damage detection threshold $T_{i}$ are obtained by using the measured strain ILs and the generated null space $N_{i,r}$ under the reference state. For the damaged state, a new index $\zeta_{i}^{*}$ is calculated using the generated null space $N_{i,r}$ together with the measured strain ILs, after which a determination is made regarding the presence of a damaged region, i.e., the region around the *i*th measured point of a quasi-static strain IL, by evaluating whether $\zeta_{i}^{*}$ is larger than the threshold $T_{i}$. The detailed damage localization procedure using the quasi-static strain IL at the *i*th measured point is illustrated schematically in Figure 2. According to the proposed method, the sensing optical fiber placed along the entire bridge superstructure possesses a high spatial resolution of measured points; thus, the damage throughout the entire bridge superstructure along the optical fiber can be estimated by repeating the above procedure using the quasi-static strain ILs at all the measured points.

## 3. Demonstration of the Effectiveness of the Proposed Method Using a Numerical Bridge Model 

Using the difference between quasi-static ILs before and after damage is a common method [19,20,21] for damage localization and detection; this approach is referred to as the traditional method in this study. In this section, a numerical model is constructed taking a continuous beam bridge as an example, and the performance of the proposed method is evaluated and compared with the traditional approach.

### 3.1. Description of the Numerical Example

A three-span continuous beam bridge composed of concrete (Young’s modulus is 3.45 × 10^7^ kN/m^3^) with a length of 60 m and a width of 16 m (5 T-shaped beams form the cross-section of the bridge) is taken as a numerical example. The whole height of each T-shape is 180 cm which consists of two parts, i.e., 40 cm for the roof thickness and another 140 cm for the height of web of each T-shape beam. The width of the roof is 280 cm and the thickness of web is 40 cm. The finite element model of this bridge is built with the software ANSYS, as shown in Figure 3. A total of 300 longitudinal elements with a length of 1 m and 244 transversal connection elements compose the whole model. As shown in Figure 3, a single optical fiber is placed on the undersides of the boundary beams along the entire three-span bridge (one beam per span) running in the longitudinal direction of the bridge. Additionally, a simulated damaged region (the #11 element in the midspan of the first span) is shown in red in Figure 3.

### 3.2. Damage Localization Performance of the Proposed Method Considering the Effects of Measurement Noise

In this three-span continuous bridge, the boundary beams to which the sensing optical fiber is attached consist of 60 elements that are numbered from #1 to #60, where elements #11, #31, and #51 are located at the midspan point of each span of the bridge. The sensing optical fiber has a 1 m spatial resolution; accordingly, a total of 60 measured points with quasi-static strain ILs are distributed along the entire bridge. For one load test, 60 quasi-static strain ILs corresponding to the 60 elements are obtained. If one of these 60 elements is damaged, the quasi-static strain ILs obtained at the measured point of this element before and after damage should theoretically be different, and the difference in the strain ILs before and after damage should be in direct proportion to the extent of damage in this element. Therefore, these 60 quasi-static strain ILs are applied to identify the presence of potential damage in the corresponding 60 elements of the three boundary beams in the longitudinal direction of the bridge. During the simulation analysis, the following cases listed in Table 1 are investigated to compare the damage localization performance between the proposed method and the traditional approach.

The effects of measurement noise on damage localization are ignored at first. Three cases, i.e., case 1, case 2, and case 5, are applied without the effects of measurement noise to compare the performance between the proposed method and the traditional approach. Herein, a 1% reduction in the element stiffness is applied to each damaged element, i.e., damaged elements #11, #31, and #51. Therefore, only a small amount of damage is considered, and it is difficult to directly detect by visual inspection.

As shown in Figure 4a, although the quasi-static strain ILs measured at the damaged elements are drawn together, it is difficult to directly identify the discrepancy in the quasi-static strain ILs before and after damage (case 2 for a single damaged element). The difference in the quasi-static strain ILs before and after damage for these two cases are drawn in Figure 4b. For each strain IL, a total of 64 positions of the moving load are considered; hence, the peak values of the curves shown in the above four figures are close to the location of damage. Therefore, it is relatively easy to identify the damaged locations in case 2 from Figure 4b. As shown in Figure 5, the damaged element is clearly and accurately identified using the proposed method. For multiple damaged elements, the similar results are obtained using above two methods, as shown in Figure 6 and Figure 7. In other words, both the proposed method and the traditional approach are effective at localizing the damage under conditions without any interference from measurement noise.

The effects of measurement noise on the localization of damage are taken into account in the following analysis. Four cases with different noise levels are applied to compare the damage localization performance between the proposed method and the traditional approach. The results of the performance comparison are shown in Figure 8 and Figure 9 (for a single damaged element) and Figure 10 and Figure 11 (for multiple damaged elements), respectively.

To simulate the interference from measurement noise, random white noise is added directly to the strain ILs. The simulation under the healthy state (case 1) is repeated with random noise nine times for each noise level; the first iteration is deemed the reference state, and the other 8 are used to determine the threshold defined in Equation (32). For the sake of brevity, only two noise levels (low and high) are presented. As shown in Figure 8b and Figure 10b, under the effects of a low level of measurement noise, the traditional method can localize the damage in the case of a single damaged element, but it is difficult to detect the locations of damage with multiple damaged elements. Figure 8d and Figure 10d clearly show that the traditional method does not work well for the situation with a high noise level. In contrast to the traditional method, the proposed approach can accurately localize the damage regardless of whether the noise level is low or high, as shown in Figure 9 and Figure 11. Theoretically, relative to the noise level on one strain IL, the noise level within the difference between two strain ILs before and after damage is enlarged. Therefore, for the traditional method, the effects of a high level of measurement noise are larger than the influence of 1% damage in both the case with a single damaged element and the case with multiple damaged elements, and thus, the traditional method fails in situations featuring the effects of a high level of measurement noise. As discussed above, the proposed method is more sensitive to damage and more robust to the effects of measurement noise than the traditional approach.

### 3.3. Damage Localization Performance of the Proposed Method Considering the Effects of Loading Conditions

All the results obtained in Section 3.2 do not consider inconsistencies in the loading conditions before and after damage; accordingly, this point is emphasized in the following analysis. As discussed in Section 2.2.1, a method that relies on the difference in the strain ILs does not work when the weight of the quasi-static loading vehicle is different before and after damage. To assess the effectiveness of the proposed method, case 1, case 8, and case 9 are taken as examples, and an inconsistency in the loading condition is achieved by using quasi-static loading vehicles with weights of 100 kN and 80 kN for the healthy and damaged bridges, respectively. The simulation under the healthy state (case 1) is repeated with different random noise (0.1%, 1.0%, 2.5% noise level respectively) 9 times for each noise level; the first iteration is deemed the reference state, and the other 8 are used to be the healthy state. Meanwhile, the healthy state without noise also is tested for the inconsistency of the weight of load vehicles before and after damage. As shown in Figure 12, an obvious difference exists between the strain ILs for the healthy bridge acted upon by the two different quasi-static loads, and thus, it is deduced that damage has occurred in the region around element #11. Unlike the traditional approach, the proposed method works effectively under inconsistent loading conditions before and after damage, as shown in Figure 13. Therefore, the proposed damage localization index defined in Equation (21) is effective even though there is a large difference in the loading conditions before and after damage. However, for practical applications of the proposed method, the weight of the loading vehicle before and after damage should be as consistent as possible and should be sufficiently heavy to obtain accurate data on the potential damage of the bridge, as described in Section 2.1.1.

## 4. Demonstration of the Effectiveness of the Proposed Method Using an Experimental Bridge Model 

### 4.1. Description of the Model Bridge

The entire experimental system in this section consists of three parts, namely, the superstructure of the simply supported model bridge, the bridge bearing and supporting frame, and the loading vehicle, as shown in Figure 14. The superstructure of the bridge is composed of three T-shaped steel beams (with a Young’s modulus of 2.0 × 10^11^ Pa), a detailed drawing of which is shown in Figure 15. Two steel bridge bearings are set at two ends of each T-shape beam. Each bridge bearing is connected to the supported steel frame using the bolts, as shown in Figure 15. Two T-shape beams are connected to the special steel connection in the transverse direction of the model bridge, and total 11 transverse connection are designed along the longitudinal direction of the model bridge. The damage is simulated by removing arbitrary transverse connection during the experiments. 

The moving vehicle system includes an electronic motor and a steel vehicle in addition to several steel clump weights (each of which weighs 20 kg) and two aluminum tracks, as shown in Figure 16. The two aluminum tracks keep the vehicle moving over the bridge along the designed trajectory, and the electronic motor moves the vehicle at a uniform slow velocity. In this study, the speed of the moving vehicle is controlled to 2 m/s to obtain the quasi-static strain ILs. 

### 4.2. Introduction of the Placement of the Sensing Optical Fiber

The sensing optical fiber is placed on the undersides of the #1 beam and the #2 beam in the longitudinal direction of the model bridge, as shown in Figure 17a,b. The differential pulse-width pair BOTDA (DPP-BOTDA) technique [42] is used to obtain a spatial resolution of 5 cm for the measured points along one sensing optical fiber. An independent research and development fiber-optical demodulator at the Harbin Institute of Technology is used to acquire the strain IL data at each measured point; the software interface and fiber-optical demodulator are shown in Figure 17c,d, respectively.

As shown in Figure 18, a total of 22 devices are designed to simulate the damage of the transverse connection between the two T-shaped beams. Different damage extents are simulated by removing different numbers of the above devices. Considering the #1 and #2 T-shaped beams supporting the moving vehicle, a total of 50 strain IL measurement points for the two beams, i.e., 25 measured points for each beam, are generated based on the 5 cm spatial resolution of measured points for one sensing optical fiber, as drawn in Figure 18.

### 4.3. Performance Comparison Between the Proposed Method and the Traditional Approach

In contrast to the numerical example described in Section 3, all the quasi-static strain ILs are classified into different clusters according to the regional distribution of measured points. As shown in Figure 18, five zones known as zone 1 through zone 5 are classified along the longitudinal direction of the bridge, and a total of ten strain IL measurement points are included in each zone. During the damage localization process, each zone is treated as one unit, e.g., the *i*th strain IL belongs to the *j*th zone; then, the damage localization index of each zone is generated with all the strain ILs belonging to zone *j*. Because the strain reflects the local deformation of the bridge superstructure, the strain ILs far from the *i*th measured point are ignored without reducing the accuracy of the damage localization. In applications to actual bridges, the number of quasi-static strain IL measurement points associated with the BOTDA technique may be quite large, and under this situation, the abovementioned zone classification should be adopted to improve the damage localization efficiency with the proposed method.

Four cases are taken as examples to compare the performance between the proposed method and the traditional approach; a detailed description of each experimental case is listed in Table 2. For experimental case 1, the simulation is repeated nine times. The first one is deemed the reference state for the sake of establishing the null space of the Hankel matrix $\Pi_{i,r}$ by using the measured strain ILs of the load test, and the others are set as the healthy state of the bridge to generate the damage localization threshold. For the damaged case, only the removal of the 17th transverse connection between the #1 beam and the #2 beam is considered because the damage in this case is considered small and is apt to demonstrate the effectiveness of the proposed method. For each damage case, the experimental load test is implemented just one time. Additionally, a loading vehicle with an inconsistent weight is implemented in experimental case 3 and experimental case 4.

The measured quasi-static strain ILs of the first five points of the #1 beam are drawn in Figure 19a, and the quasi-static strain ILs at the 13th point of the #1 beam before and after damage are compared in Figure 19b. It is difficult to directly differentiate the measured quasi-static strain ILs between the healthy state (experimental case 1) and the damaged state (experimental case 2). With the traditional method, the difference between the quasi-static strain ILs before and after damage at the 13th point of the #1 beam is drawn in Figure 19c; the 13th measured point is selected because this point is the nearest point to the damaged location, i.e., the 17th transverse connection between the #1 beam and the #2 beam. Evidently, the traditional method does not work for this damaged case. This failure originates from the fact that the quasi-static strain ILs no longer follow a smooth theoretical curve but instead exhibit fluctuations, and thus, the difference in the ILs is no longer sensitive to the damage of the bridge, especially under conditions of small damage. The damage localization results with the proposed approach are shown in Figure 20. In contrast to the results of the traditional method, it appears that the damage localization index values of the measured points near the midspans of the #1 beam and #2 beam, e.g., the 13th point of the #1 beam and the 13th point of the #2 beam, are larger than the threshold value. The other four strain ILs near these two measured points also show relatively large damage localization index values because these measured points are all close to the region with the damaged transverse connection between the two beams. Therefore, using this experimental example, the effectiveness of the proposed method is demonstrated through a comparison with the results of the traditional approach.

The data obtained from experimental cases 1, 3 and 4 are used here to evaluate the performance of the proposed method and the traditional approach based on considerations of the differences in the loading conditions before and after damage. With the traditional method, as shown in Figure 21b, there is a clear and large difference in the quasi-static strain ILs between experimental case 1 and experimental case 3 (the 13th measured point of the #1 beam). Therefore, although the model bridge is in a healthy state in both cases, an incorrect conclusion of damage is drawn, and thus, the traditional method is invalid for situations involving different loading conditions before and after damage. The results illustrated in Figure 22 using the proposed method show that damage can be localized even when the loading conditions before and after damage are different; moreover, an erroneous determination of damage is avoided for both healthy states represented by experimental case 1 and experimental case 3. The above discussion demonstrates that the proposed method is more effective and robust than the traditional approach at damage localization based on quasi-static strain ILs.

## 5. Conclusions

In this study, a method is proposed to localize damage in the superstructure of beam bridges by using measured quasi-static strain ILs based on the BOTDA technique. The following conclusions are drawn:

(i) The BOTDA technique is first applied to localize damage based on the quasi-static strain ILs of beam bridges, and thus, the scarcity of strain IL measurement points is improved greatly.

(ii) The energy ratio between the main components of all strain ILs and the measurement noise is effective to mitigate the effects of measurement noise on the damage localization results for bridges. 

(iii) The proposed damage localization index based on the subspace between quasi-static strain ILs is sensitive to damage in the bridge superstructure and insensitive to differences in the loading conditions before and after damage. 

(iv) In comparison with the traditional method based on the difference between the strain ILs before and after damage, the proposed method is more sensitive to the damage location and more robust in cases with measurement noise.

## Figures and Tables

**Figure 1 sensors-18-04446-f001:**
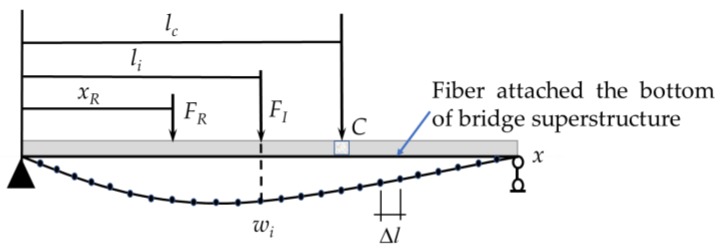
Schematic of the quasi-static loading of a bridge with a long sensing fiber.

**Figure 2 sensors-18-04446-f002:**
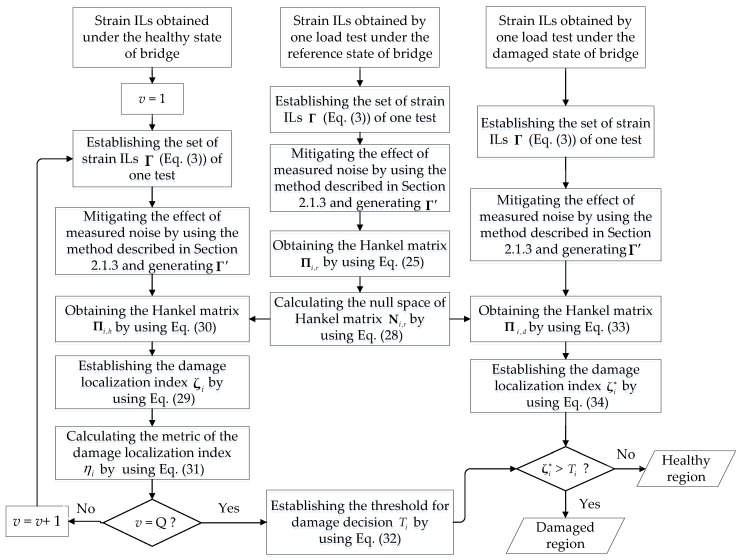
Diagram of the damage localization method (for the strain IL at the *i*th measured point).

**Figure 3 sensors-18-04446-f003:**
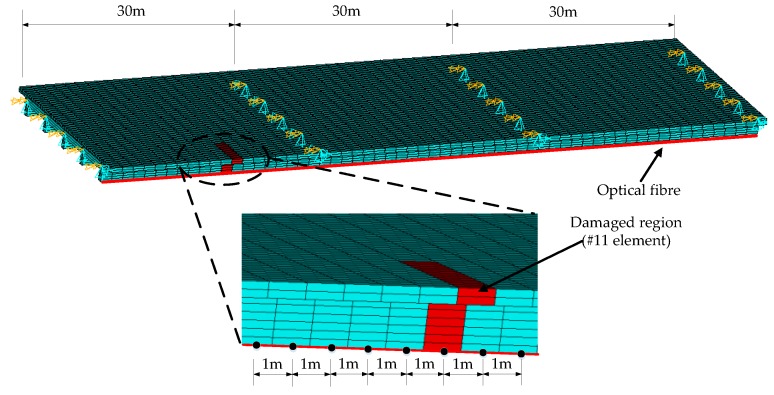
Schematic of the quasi-static loading of a bridge with a long sensing fiber.

**Figure 4 sensors-18-04446-f004:**
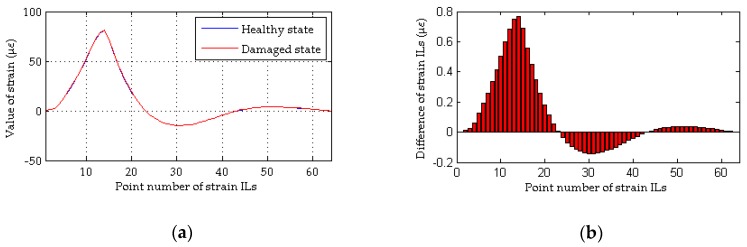
Results of the traditional method without the effects of measurement noise (for a single damaged element): (**a**) strain ILs at point #11 for case 1 and case 2; (**b**) difference in the strain ILs at point #11 before and after damage for case 2.

**Figure 5 sensors-18-04446-f005:**
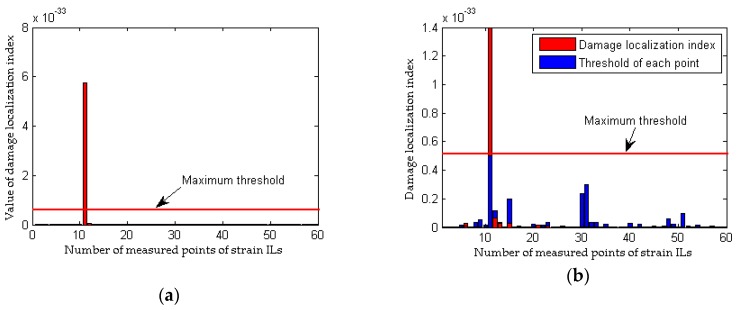
Results of the proposed method without the effects of measurement noise (for a single damaged element): (**a**) results of the proposed method for case 2; (**b**) detailed magnification of the results of the proposed method for case 2.

**Figure 6 sensors-18-04446-f006:**
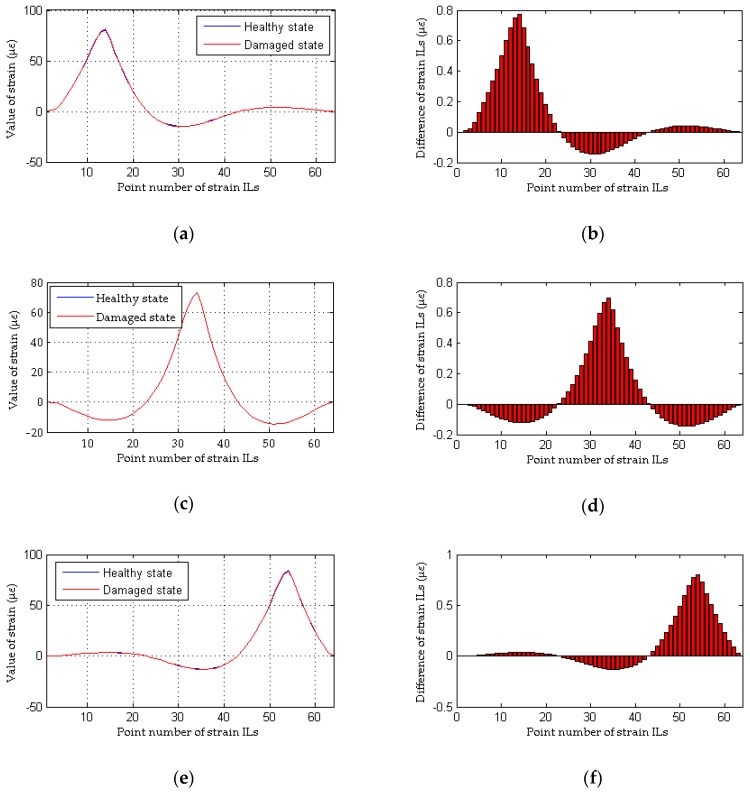
Results of the traditional method without the effects of measurement noise (for multiple damaged elements): (**a**) strain ILs at point #11 for case 1 and case 5; (**b**) difference in the strain ILs at point #11 before and after damage for case 5; (**c**) strain ILs at point #31 for case 1 and case 5; (**d**) difference in the strain ILs at point #31 before and after damage for case 5; (**e**) strain ILs at point #51 for case 1 and case 5; (**f**) difference in the strain ILs at point #51 before and after damage for case 5.

**Figure 7 sensors-18-04446-f007:**
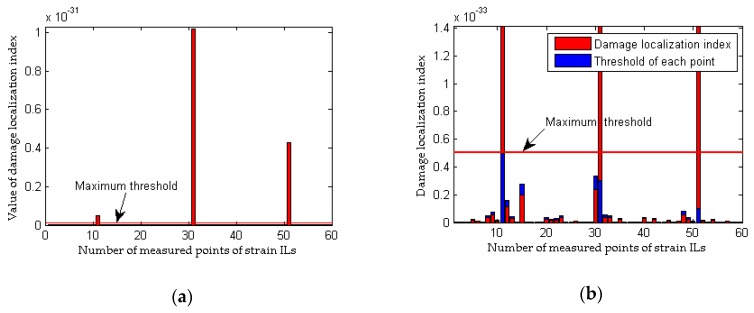
Results of the proposed method without the effects of measurement noise (for multiple damaged elements): (**a**) results of the proposed method for case 5; (**b**) detailed magnification of the results of the proposed method for case 5.

**Figure 8 sensors-18-04446-f008:**
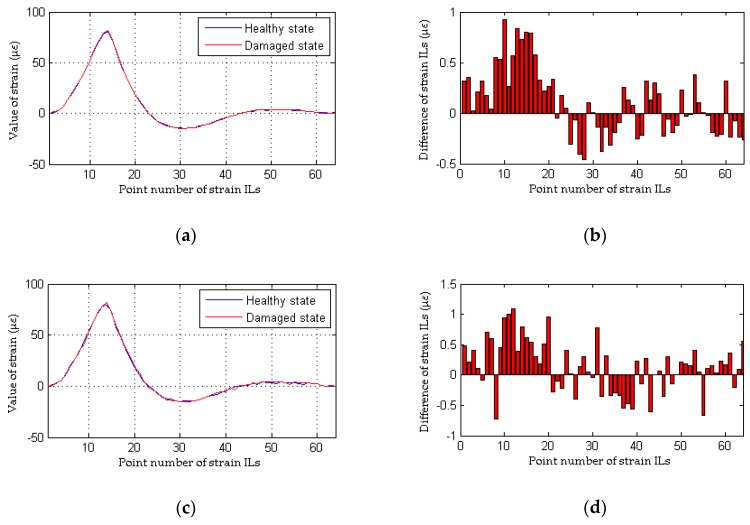
Results of the traditional method considering the effects of measurement noise (for a single damaged elements): (**a**) strain ILs at point #11 for case 1 and case 3; (**b**) difference in the strain ILs at point #11 before and after damage for case 3; (**c**) strain ILs at point #11 for case 1 and case 4; (**d**) difference in the strain ILs at point #11 before and after damage for case 4.

**Figure 9 sensors-18-04446-f009:**
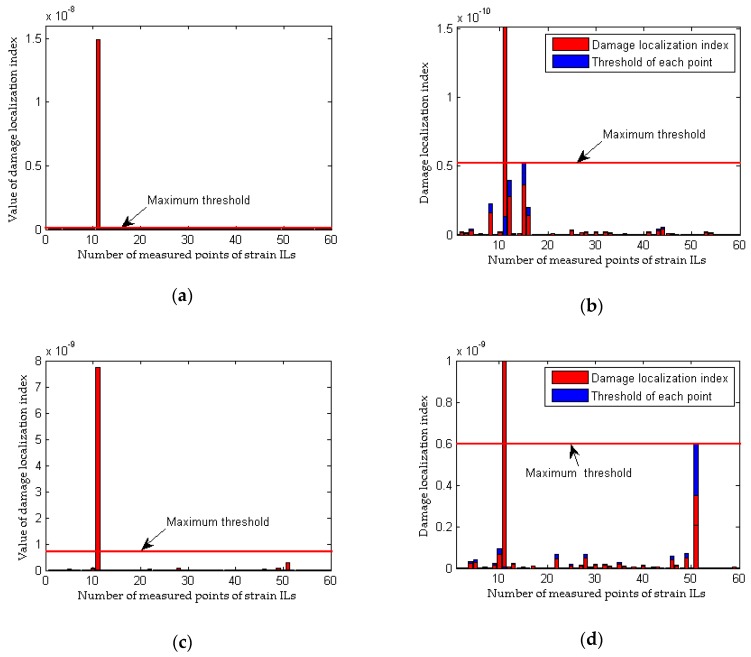
Results of the proposed method considering the effects of measurement noise (for a single damaged elements): (**a**) results of the proposed method for case 3; (**b**) detailed magnification of the results of the proposed method for case 3; (**c**) results of the proposed method for case 4; (**d**) detailed magnification of the results of the proposed method for case 4.

**Figure 10 sensors-18-04446-f010:**
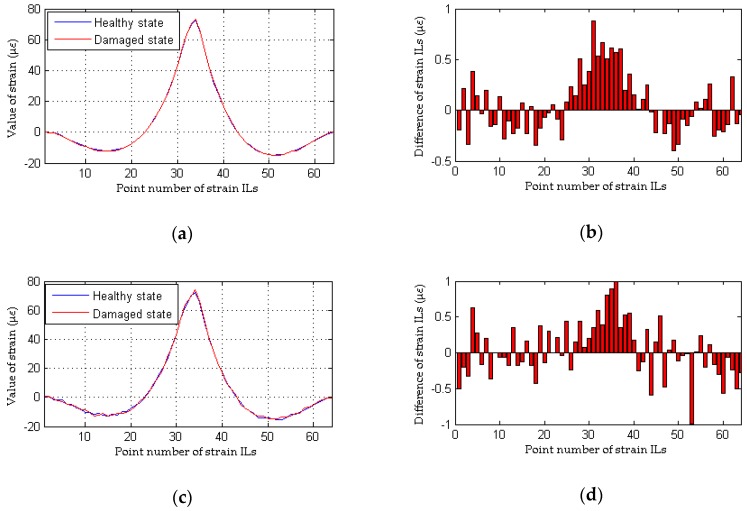
Results of the traditional method considering the effects of measurement noise (for multiple damaged elements): (**a**) strain ILs at point #31 for case 1 and case 6; (**b**) difference in the strain ILs at point #11 before and after damage for case 6; (**c**) strain ILs at point #31 for case 1 and case 7; (**d**) difference in the strain ILs at point #31 before and after damage for case 7.

**Figure 11 sensors-18-04446-f011:**
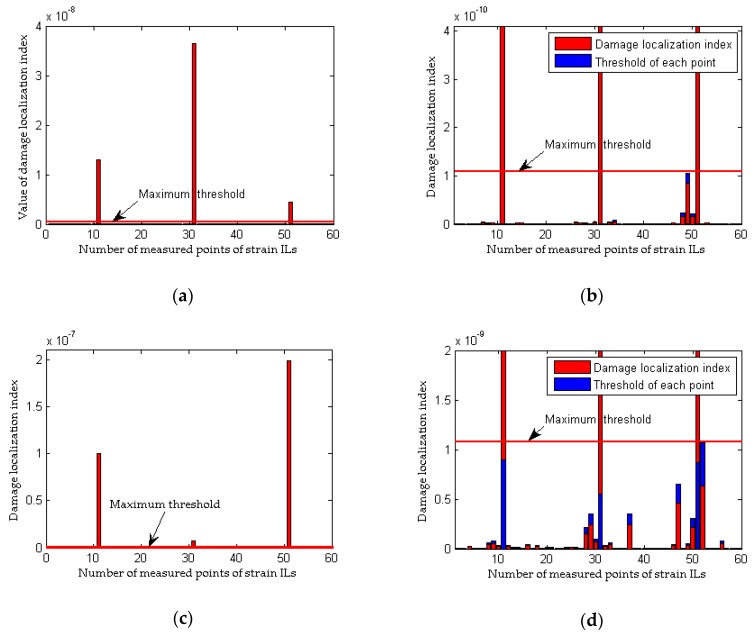
Results of the proposed method considering the effects of measurement noise (for multiple damaged elements): (**a**) results of the proposed method for case 6; (**b**) detailed magnification of the results of the proposed method for case 6; (**c**) results of the proposed method for case 7; (**d**) detailed magnification of the results of the proposed method for case 7.

**Figure 12 sensors-18-04446-f012:**
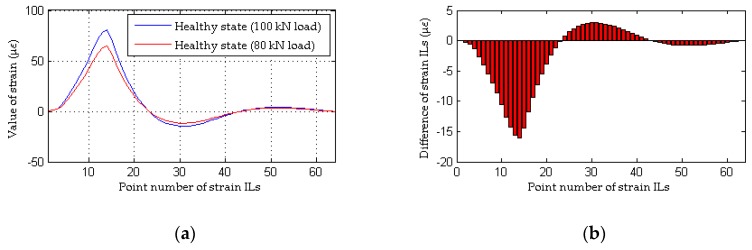
Results of the traditional method considering the effects of loading conditions: (**a**) strain ILs at point #11 between case 1 and case 8; (**b**) difference in the strain ILs at point #11 before and after damage for case 8.

**Figure 13 sensors-18-04446-f013:**
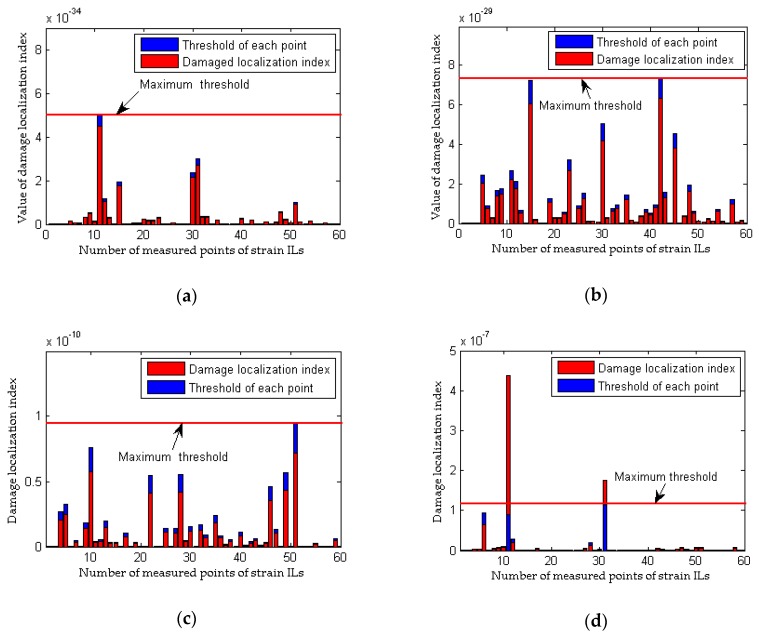
Results of the proposed method considering the effects of loading conditions: (**a**) results of the proposed method for case 8 without noise; (**b**) results of the proposed method for case 8 with 0.1% noise level; (**c**) results of the proposed method for case 8 with 1.0% noise level; (**d**) results of the proposed method for case 9 with 2.5% noise level.

**Figure 14 sensors-18-04446-f014:**
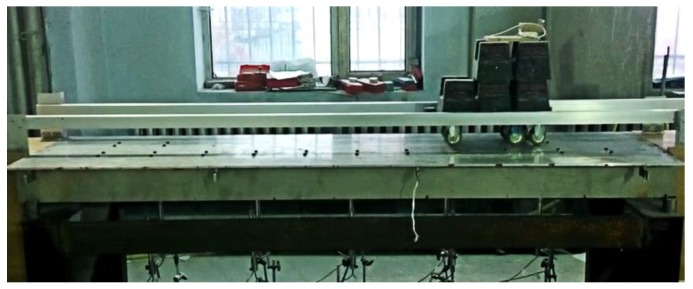
Photo of the entire experimental system of the model bridge.

**Figure 15 sensors-18-04446-f015:**
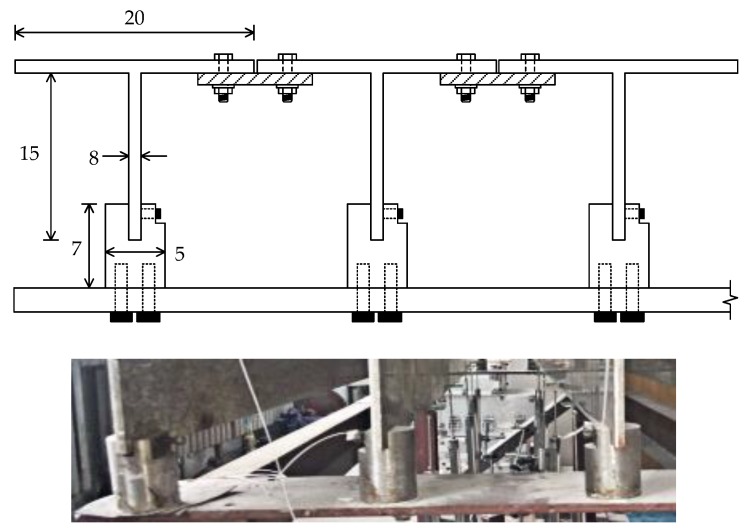
Detailed description and photo of a cross-section of the model bridge (unit: mm).

**Figure 16 sensors-18-04446-f016:**
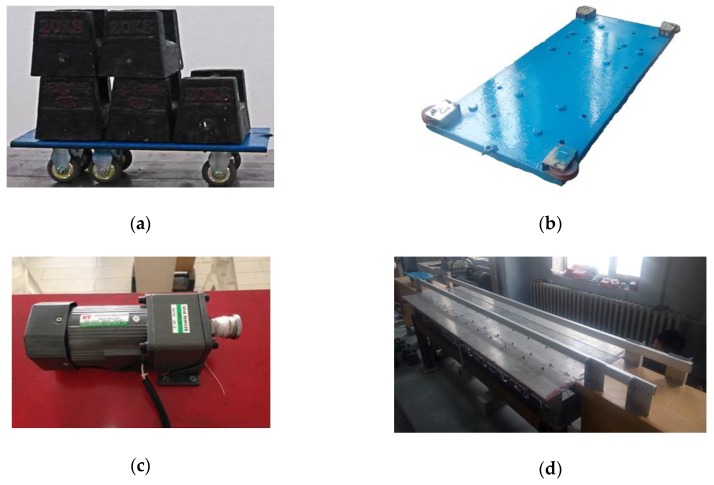
Photographs of the moving vehicle system: (**a**) clump weights; (**b**) steel vehicle; (**c**) electronic motor; (**d**) two aluminum tracks.

**Figure 17 sensors-18-04446-f017:**
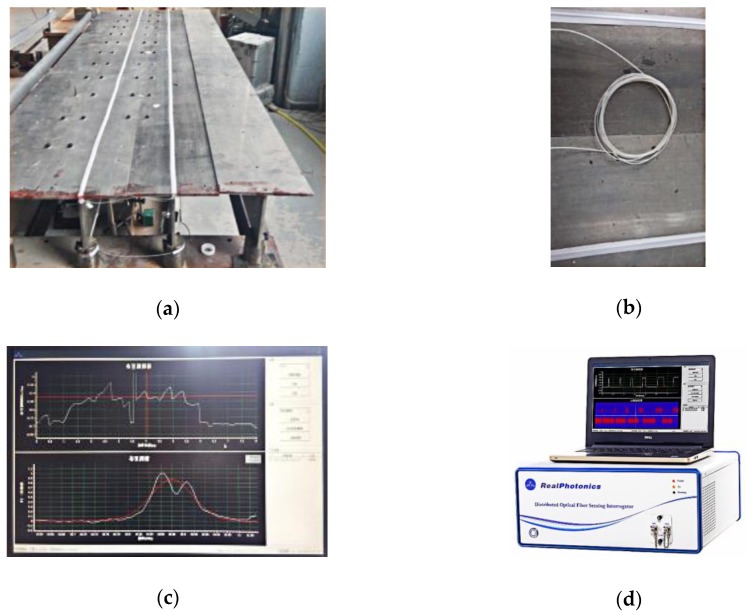
Photographs of the sensing system: (**a**) placement of the optical fiber; (**b**) optical fiber; (**c**) software interface of the fiber optic demodulator; (**d**) fiber-optical demodulator.

**Figure 18 sensors-18-04446-f018:**
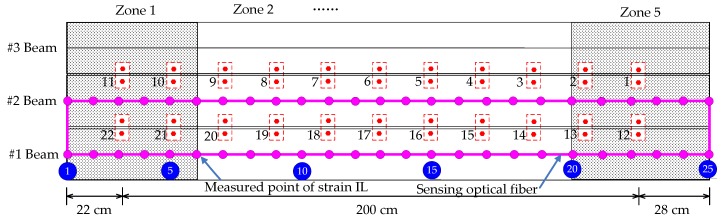
Placement of the sensing optical fiber and devices simulating the damage of the transverse connection.

**Figure 19 sensors-18-04446-f019:**
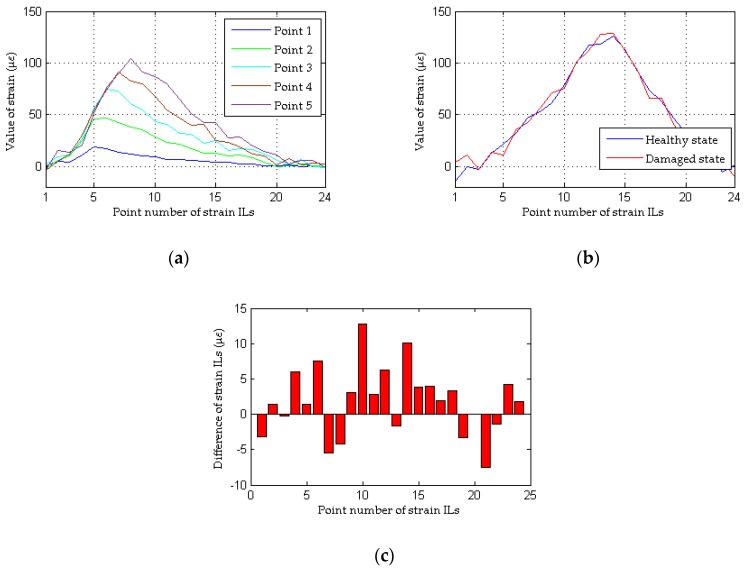
Results of the traditional method for the model bridge: (**a**) strain ILs of the first five measured points of the #1 beam for experimental case 1; (**b**) strain ILs at point #13 of the #1 beam for experimental case 1 and experimental case 2; (**c**) difference in the strain ILs at point #13 before and after damage for experimental case 2.

**Figure 20 sensors-18-04446-f020:**
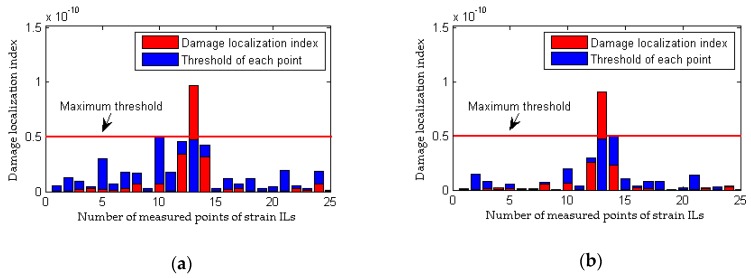
Results of the proposed method for the model bridge: (**a**) results of the #1 beam for experimental case 2; (**b**) results of the #2 beam for experimental case 2.

**Figure 21 sensors-18-04446-f021:**
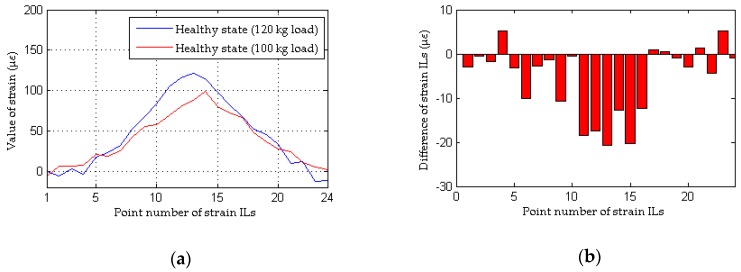
Results of the traditional method for the model bridge considering the effect of loading condition: (**a**) strain ILs at point #13 of the #1 beam for experimental case 1 and experimental case 3; (**b**) difference in the strain ILs at point #13 between before and after damage for experimental case 3.

**Figure 22 sensors-18-04446-f022:**
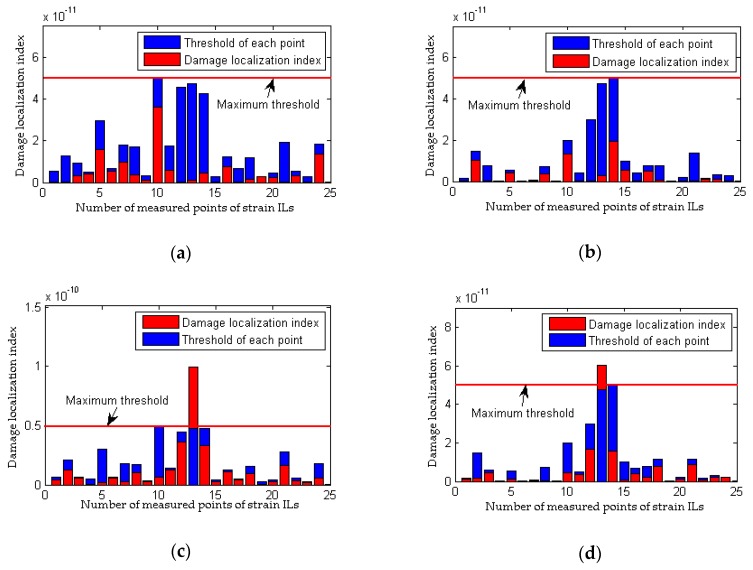
Results of the proposed method for the model bridge considering the effect of loading condition: (**a**) results of the #1 beam for experimental case 3; (**b**) results of the #2 beam for experimental case 3; (**c**) results of the #1 beam for experimental case 4; (**d**) results of the #2 beam for experimental case 4.

**Table 1 sensors-18-04446-t001:** Descriptions of all cases for the numerical example.

Case Number	Description of Case	Case Number	Description of Case
Case 1	Healthy bridge (100 kN quasi-static moving load)	Case 6	#11, #31 and #51 damaged elements with 0.5% noise
Case 2	#11 damaged element ^1^ without noise	Case 7	#11, #31 and #51 damaged elements with 2.0% noise
Case 3	#11 damaged element with 0.5% noise	Case 8	Healthy bridge (80 kN quasi-static moving load)
Case 4	#11 damaged element with 2.0% noise	Case 9	#11 and #31 damaged elements with 2.5% noise (80 kN quasi-static moving load)
Case 5	#11, #31 and #51 damaged elements without noise		

^1^ The extent of damage of each damaged element in each case is simulated as a 1% reduction in the stiffness.

**Table 2 sensors-18-04446-t002:** Descriptions of all cases for the experimental example.

Case Number	Description of Case	Case Number	Description of Case
Experimental case 1	Healthy bridge (120 kg moving load)	Experimental case 3	Healthy bridge (100 kg moving load)
Experimental case 2	Removing the #17 transverse connection (120 kg moving load)	Experimental case 4	Removing the #17 transverse connection (100 kg moving load)

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
