# Peer review of "Damage Localization of Beam Bridges Using Quasi-Static Strain Influence Lines Based on the BOTDA Technique"

_sensors, 2018, doi:10.3390/s18124446_

Reviewer 1 Report

The paper deals with the diagnosis of damage in bridge superstructures using quasi-static strain influence lines (ILs) using the Brillouin optical time domain analysis (BOTDA) technique. The manuscript provides a particularly detailed explanation of the theory behind the analysis.

My main concern is that the manuscript is sheer length in some sections (e.g. theoretical background) obscures the advances in scientific knowledge contained within. Some relevant information to understand the context of the study are omitted. A thorough review to point out the big lessons learned from the study in view of more general recommendations is mandatory.

The main weaknesses of this manuscript are as follows:

1) The novelty of the research and recommendations need to be transparent.

2) There is lack of referencing throughout the script is noted. The results are not compared to relevant research in the area of damage detection using influence lines.

Others:

1)  It is stated that “multiple types of damage information across the entire bridge superstructure” can be detected using this technique. Needs to be specified.

2) Each variable and constant need explanation and units.

3) The authors suggest to use approximately 0.99 cut value in order to retain the majority of the information on all measured quasi-static strain ILs of bridges. Since, this cut value is noise level dependant how it is determined? The statement need to be referenced and discussed.

4) The explanation of damage localisation threshold choice is needed with the discussion and parallel comparison of results shown, e.g. undamaged, damaged, with and without noise. Different cut off used throughout the figures.

5) The experiment deals only with the healthy state of the beams. How the damage localisation I practice will work using this technique? How the optic sensors will be applied to full structure and data analysed? What is the processing time? 

6) Discuss more on disadvantages of the technique, e.g. contact measurements needed, closing the bridge for collecting data, etc.

Author Response

We do appreciate your comments and advices. We carefully discussed the your advices and comments, and the detailed response and modifications to each question are shown as follows. Please check them.

Reviewer 2 Report

The paper deals with the development of novel damage detection technique using BOTDA type of FO sensors. The method proposes use of Influence lines, determines thresholds and overcomes measurement noise. The paper is organized well with description of the technique followed by numerical and experimental validation.

The reviewer believes that the paper needs the following changes to make it worthy for publication in an international journal

1)      A complete discussion about the state of the art, including citations of papers from the last 5 years.

2)      The discussion about the source of other uncertainties and the effects that it might have on the success of the method. For instance the temperature effects have been completely neglected. Only one sentence saying that the measurements for the damaged healthy structure need to be carried out when the ambient conditions are same, but unfortunately it is not always possible. So how can this be overcome? Experimental and numerical studies may need to be carried out for that.

Soman et al. have carried out numerical assessment of temperature effects both bulk and gradient. The authors are guided to those papers as reference

3)      The details of the experiment need to be provided, the speed of the vehicle and the reason behind the clubbing of the bridge sections needs to be explained and how the methodology will be implemented in the real applications. If the damage detection can be achieved only to determine the section.

4)      The robustness of the method to higher levels of noise needs to be carried out. Upto 10% noise may be expected in the strain measurements.

5)      The scale of the figure 7c is 10^-29 which is very small and there are concerns about the computational accuracy and the round-off errors

6)      The photos included in the paper are not very clear and some improvements need to be introduced.

Apart from these major concerns there are some minor typos and grammar mistakes in the paper.

More details of the numerical model need to be provided as well.

Author Response

We do appreciate your comments and advices. We carefully discussed the your advices and comments, and the detailed response and modifications to each question are shown as follows. Please check them.

Round  2

Reviewer 2 Report

The paper is significant improvement over the first version. But some sections require more work before the paper can be considered good enough for publication in an international journal.

The additions made in lines 102-113 about the ambient conditions are not convincing. either they need to be improved or the test runs for the temperature uncertainty should be investigated.

The discussion in section 3.3 should also include comparison of IL's with 1-% noise to determine the possibility of false negative.

Also the discussions on the use of denoising should be expanded. A sample section iwth only the denoising for healthy scenarios may be added as well.

The answer to question 5 from the previous review is not satisfactory. A very small damage index leads to low confidence in the damage detection. Also in the eyes of the reviewer if the strain values are 50-100 micro strains and the damage index is 10^-29, the chances of damage detection appear to be low. so more convincing answer needs to be provided.

Lastly, there are some typos still in the paper

line 168, for instance there is grammar mistake. Also in line 26, what do the authors mean by strain modal? 

The reviewer might also point to some numerical work carried out by Soman et al. which might be interesting for the readers.

Soman R., Kyriakides M, Onoufriou T., Ostachowicz W.: Numerical Evaluation of Multi-Metric

Data Fusion based Structural Health Monitoring of Long Span Bridge Structures. Journal of Structure and Infrastructure Engineering
